# Revisiting DP-Means: Fast Scalable Algorithms
# via Parallelism and Delayed Cluster Creation

**Or Dinari**[1]                                    **Oren Freifeld**[1]

[1]The Department of Computer Science, Ben-Gurion University of the Negev, Be'er Sheva, Israel

## Abstract

DP-means, a nonparametric generalization of K-means, extends the latter to the case where the number of clusters is unknown. Unlike K-means, however, DP-means is hard to parallelize, a limitation hindering its usage in large-scale tasks. This work bridges this practicality gap by rendering the DP-means approach a viable, fast, and highly-scalable solution. First, we study the strengths and weaknesses of previous attempts to parallelize the DP-means algorithm. Next, we propose a new parallel algorithm, called PDC-DP-Means (Parallel Delayed Cluster DP-Means), based in part on delayed creation of clusters. Compared with DP-Means, PDC-DP-Means provides not only a major speedup but also performance gains. Finally, we propose two extensions of PDC-DP-Means. The first combines it with an existing method, leading to further speedups. The second extends PDC-DP-Means to a Mini-Batch setting (with an optional support for an online mode), allowing for another major speedup. We verify the utility of the proposed methods on multiple datasets. We also show that the proposed methods outperform other nonparametric methods (*e.g.*, DBSCAN). Our highly-efficient code can be used to reproduce our experiments and is available at https://github.com/BGU-CS-VIL/pdc-dp-means.

## 1 INTRODUCTION

In the age of "Big Data", algorithms that scale poorly, even if they offer desiderata of useful properties, are often discarded in favor of faster and more scalable alternatives. This phenomenon is exemplified, in clustering tasks, by the wide popularity of the simple K-Means algorithm. One main reason for that popularity is that, by the virtue of the ease in

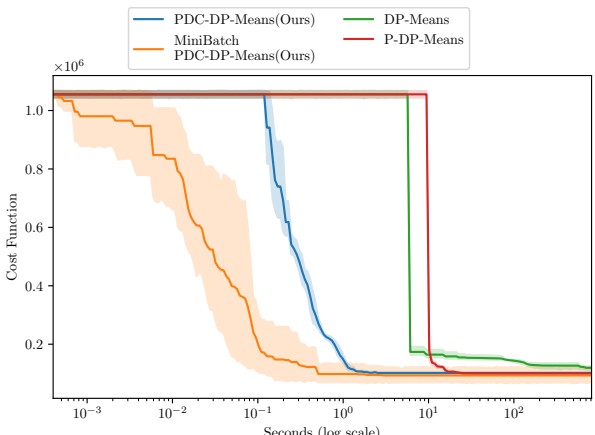

Figure 1: Convergence of the different algorithms (note the logarithmic scale of the abscissa). MiniBatch PDC-DP-Means converges before the completion of the first iteration of PDC-DP-Means, which in turn converges before the completion of the first iteration of either DP-Means or P-DP-Means. Data: 1 million 2D points generated from 50 Gaussians. Results are average (solid lines) ± std. dev. (shaded areas) of 5 runs. See § 5 for details.

which K-Means lends itself to parallelization and optimized computations, its speed is unrivaled by most other methods. Thus, in large-scale clustering tasks, K-Means is often the weapon of choice.

In particular, in practice K-Means is usually preferred over the more powerful and elegant DP-Means algorithm [Kulis and Jordan, 2012] (and its variants). This is despite the fact that with DP-Means, the user obtains K-Means-like clustering with the added benefit of being free from having to specify (or guess) the value of $K$, the number of clusters.

Although there are numerous cases where, at least in theory, it would have made sense to use DP-Means instead of K-Means, DP-Means has a major drawback that hinders its applicability: at least in its original formulation, DP-Means

*Accepted for the 38th Conference on Uncertainty in Artificial Intelligence* (UAI 2022).

cannot be parallelized as efficiently as K-Means. Bridging this practical gap is the topic of this paper (see Figure 1).

We start by studying two existing methods that attempt to parallelize DP-Means: **DACE** [Jiang et al., 2017] and Parallel DP-Means (**P-DP-Means**) [Pan et al., 2013]. This study then leads us to understand not only what makes DP-Means hard to parallelize but also how to overcome this difficulty. Based on those insights, we propose new algorithms, all targeting the minimization of the DP-Means cost function. The first proposed algorithm, on top of which the others are built, is Delayed Cluster DP-Means (**DC-DP-Means**). This *serial* algorithm is a new variant of the original (and also serial) DP-Means algorithm. DC-DP-Means has two important advantages over the vanilla DP-means: 1) it is less prone to over-clustering and usually achieves better clustering results; 2) it removes the limitation which halted K-Means-like parallelization in DP-Means. That second advantage lets us propose our first parallel algorithm, Parallel DC-DP-Means (**PDC-DP-Means**), which lends itself, by design, to an extremely-efficient implementation which *rivals the speed of popular and optimized K-Means implementations* while having the same performance as DC-DP-Means.

Next, we extend PDC-DP-Means to *two additional parallel algorithms*. The first, **DACE-PDC- DP-Means**, uses PDC-DP-Means in conjunction with DACE, achieving greater speed than either DACE or PDC-DP-Means. The second, **MiniBatch PDC-DP-Means**, is for a Mini-Batch setting, can be used in either an online or an offline mode, and offers an additional major speedup over our already-fast PDC-DP-Means. To put the speedup in perspective, our Mini-Batch PDC-DP-Means clusters the entirety of ImageNet's [Deng et al., 2009] train-set (following dimensionality reduction, via feature extraction, to 128) in as little as 13 seconds. *This is orders of magnitude faster than the competitors* (who were given the same 128-dimensional features as input). The full comparison and details for that experiment appear in § 5. Importantly: 1) such a large-scale task is outside the scope of the original DP-Means; 2) while previous parallel DP-Means methods can handle such large data, we do it in a fraction of the time it takes them. More generally, we are unaware of any nonparametric method (including ones unrelated to DP-Means) that is even close to that speed.

To summarize, our main contributions are: 1) recognizing what stymied previous DP-Means parallelization methods; 2) we propose the Delayed Cluster DP-Means algorithm, and its parallel version, the PDC-DP-Means, offering a performance gain and a major speedup over DP-Means. 3) we propose two extensions of PDC-DP-Means (DACE-PDC-DP-Means and MiniBatch PDC-DP-Means) that offer additional major speedups.

## 2  RELATED WORK

**K-Means.** The K-Means cost function is the sum of squared $\ell_2$ norms of the residuals between the observations and the mean of the cluster each of them is assigned to; *i.e.*,

$$\sum_{k=1}^{K} \sum_{i:z_i=k} \|\boldsymbol{x}_i - \boldsymbol{\mu}_k\|_{\ell_2}^2 \qquad (1)$$

where $N$ is the number of points (or observations), $K$ is the number of clusters, $\boldsymbol{x}_i \in \mathbb{R}^d$ is data point $i$, $\boldsymbol{\mu}_k \in \mathbb{R}^d$ is the mean (or center) of cluster $k$, and the label $z_i = k$ if and only $\boldsymbol{x}_i$ belongs to cluster $k$. Thus, cluster $k$, denoted by $\mathcal{C}_k$, consists of the points whose assignment is $k$: $\mathcal{C}_k = (\boldsymbol{x}_i)_{i:z_i=k}$. The function is to be minimized w.r.t. both $(\boldsymbol{\mu}_k)_{k=1}^{K}$ and $(z_i)_{i=1}^{N}$. Importantly, the user must specify $K$. Since initially proposed by Forgy [1965], many variations of K-Means have been developed. Today, the popular approaches for performing K-Means inference are either Lloyd's algorithm [Lloyd, 1982] (usually referred to as the K-Means *algorithm*) or Elkan's [Elkan, 2003]. Both are easy to parallel, thereby offering a very fast clustering method. We remark that in this work we borrow from Lloyd [1982] several ideas related to parallelization and mini-batches.

**DP-Means.** Proposed by Kulis and Jordan [2012], the DP-Means is a nonparametric extension of K-Means, rooted in Bayesian nonparametrics, and closely-related to the Dirichlet Process Mixture model [Antoniak, 1974, West and Escobar, 1993]. In the DP-Means algorithm, when an observation's squared distance from the mean of its closest cluster exceeds a user-defined parameter $\lambda$ ($\lambda > 0$), a new cluster is formed, and the observation is assigned to it. The associated cost function is similar to K-Means, except that there is an added penalty term and the minimization is also w.r.t. $K$:

$$\left(\sum_{k=1}^{K} \sum_{i:z_i=k} \|\boldsymbol{x}_i - \boldsymbol{\mu}_k\|_{\ell_2}^2\right) + \lambda K \,. \qquad (2)$$

Note that the penalty term, $\lambda K$, penalizes the creation of new clusters. As we will explain later, the DP-Means algorithm is serial and thus is inherently slow.

Several works have extended the original DP-Means algorithm. Bachem et al. [2015] use corsets to achieve fast approximated inference; *i.e.*, the entire dataset is efficiently summarized by a small weighted subset of representative points. That approach allows to use slow algorithms, such as the original DP-Means, on large datasets. Odashima et al. [2016] proposed several DP-Means-related algorithms. First, they developed an Online DP-Means and a Batch DP-Means algorithms, both based on the MiniBatch K-Means [Sculley, 2010]. In addition, they have developed Split-DP-Means and Merge-DP-Means, and then combined them to a Split-Merge DP-Means. The splits/merges moves are used in order to try to escape poor local minima (for other Bayesian nonparametric clustering methods using splits and merges,

see, *e.g.*, Jain and Neal [2004], Chang and Fisher III [2013, 2014], Dinari and Freifeld [2020], Ronen et al. [2022]). Kobayashi and Watanabe [2021] add another term to the DP-Means cost function, making it more robust to outliers. Paul and Das [2020] proposed the EWDP-Means, which incorporates optimal feature weighting using Gibbs sampling. While all the DP-Means methods above are serial, two works which are of particular interest to us are the Parallel DP-Means, proposed by Pan et al. [2013], and DACE, proposed by Jiang et al. [2017]. Both these methods use parallelization and we will return to them later in this paper.

## 3 BACKGROUND

Below we review several well-known facts (of interest in our context) about the K-Means and DP-Means algorithms.

**Optimizing Lloyd's Algorithm.** Lloyd's K-Means algorithm (Algorithm 1) is simple enough so it can be both optimized (in terms of efficiency of the computations) and parallelized such that its speed is virtually unrivaled. In terms of running time, the most expensive part is calculating the distance between each data point and each of the $K$ cluster means. To optimize this part, it is better to use matrix multiplication than calculating such distances one by one; *e.g.*, with matrix multiplication the user can leverage Basic Linear Algebra Subprograms (BLAS) [Lawson et al., 1979] as much as possible. For that aim, note first that the (squared) distance calculation has three parts:

$$||\boldsymbol{x}_i - \boldsymbol{\mu}_k||_{\ell_2}^2 = ||\boldsymbol{x}_i||_{\ell_2}^2 - 2\boldsymbol{x}_i^T\boldsymbol{\mu}_k + ||\boldsymbol{\mu}_k||_{\ell_2}^2 . \quad (3)$$

In the RHS of Eq. (3), the first term, $||\boldsymbol{x}_i||_{\ell_2}^2$, is constant w.r.t. $k$ while the last term, $||\boldsymbol{\mu}_k||_{\ell_2}^2$, can be computed just once per iteration (instead of $N$ times per iteration). It follows that the main effort lies with computing the middle term, $2\boldsymbol{x}_i^T\boldsymbol{\mu}_k$. To make that computation efficient two things are done. First, $\boldsymbol{X}$ is split into $P$ parts, $(\boldsymbol{X}_p)_{p=1}^P$, which can be processed in parallel. Let $N_p$ denote the number of points in part $p$, and let $\boldsymbol{X}_p = \begin{bmatrix} \boldsymbol{x}_{1,p} & \dots & \boldsymbol{x}_{N_p,p} \end{bmatrix}^T \in \mathbb{R}^{N_p \times d}$ denote that part, written as a matrix. Second, instead of computing $N_p \times K$ individual vector-vector multiplication computations of $\boldsymbol{x}_{j,p}^T\boldsymbol{\mu}_k$ (one for each $(j,k)$ pair), a single matrix-matrix multiplication is performed: $\boldsymbol{X}_p\boldsymbol{M}^T$ where $\boldsymbol{M} = \begin{bmatrix} \boldsymbol{\mu}_1, \dots, \boldsymbol{\mu}_K \end{bmatrix}^T$. The above steps are the root reasons for the speedup that Lloyd's algorithm achieves over a naive implementation. However, it can be optimized even further: once each part is processed, it can produce its relative contributions to the cluster means, which in turn can be aggregated in the main process before the next iteration.

    **The DP-Means algorithm.** There are obvious similarities between the K-Means algorithm (Algorithm 1) and the DP-Means algorithm (Algorithm 2). However, while in the former $K$ is assumed to be known and is predefined, in the latter $K$ evolves during the algorithm's run and depends on:

---

**Algorithm 1:** Lloyd's K-Means Algorithm [Lloyd, 1982]

**Input:** $K$
**Data:** $\boldsymbol{X} = (\boldsymbol{x}_i)_{i=1}^N \subset \mathbb{R}^d$
1   $(\boldsymbol{\mu}_k)_{k=1}^K \leftarrow K$ randomly-chosen points from $\boldsymbol{X}$
2 **while** *Not Converged* **do**
3     **for** $i \in \{1, \dots, N\}$ **do**
4        $z_i \leftarrow \arg\min_{k \in \{1,\dots,K\}} ||\boldsymbol{x}_i - \boldsymbol{\mu}_k||_{\ell_2}^2$
5     **for** $k = 1 \in \{1, \dots, K\}$ **do**
6        $n_k \leftarrow |\{i : z_i = k\}|$
7        $\boldsymbol{\mu}_k \leftarrow \frac{\sum_{i:z_i=k} \boldsymbol{x}_i}{n_k}$

---

**Algorithm 2:** DP-Means [Kulis and Jordan, 2012]

**Input:** $\lambda$
**Data:** $\boldsymbol{X} = (\boldsymbol{x}_i)_{i=1}^N \subset \mathbb{R}^d$
1   $K \leftarrow 1$
2   $\boldsymbol{\mu}_1 \leftarrow \frac{\sum_{i=1}^N \boldsymbol{x}_i}{N}$
3   $(z_i)_{i=1}^N \leftarrow 1$      // init. all labels to 1
4 **while** *Not Converged* **do**
5     **for** $i \in \{1, \dots, N\}$ **do**
6        $z_i \leftarrow \arg\min_{k \in \{1,\dots,K\}} ||\boldsymbol{x}_i - \boldsymbol{\mu}_k||_{\ell_2}^2$
7        **if** $||\boldsymbol{x}_i - \boldsymbol{\mu}_{z_i}||_{\ell_2}^2 > \lambda$ **then**
8           $K \leftarrow K + 1$
9           $\boldsymbol{\mu}_K \leftarrow \boldsymbol{x}_i$
10          $z_i \leftarrow K$
11     **for** $k = 1 \in \{1, \dots, K\}$ **do**
12        $n_k \leftarrow |\{i : z_i = k\}|$
13        $\boldsymbol{\mu}_k \leftarrow \frac{\sum_{i:z_i=k} \boldsymbol{x}_i}{n_k}$

---

1) the data; 2) $\lambda$; 3) the ordering in which one visits the observations. When assessing the convergence of Algorithm 2, one can see that apart from the cluster creation (*e.g.* lines 7-10), it has the same guarantees as the classical K-Means (meaning, every step in the algorithm cannot increase the cost, which is bounded below by zero). When taking into account the addition of clusters, and examining the DP-Means cost function, it can be noted that adding a cluster is done only when all the squared distances between some observation $\boldsymbol{x}_i$ and each of the $K$ cluster means exceed $\lambda$. Thus, the penalty term (*i.e.*, $\lambda K$) is smaller than the squared distance between $\boldsymbol{x}_i$ and any of the existing $K$ clusters. Thus, the creation of the cluster necessarily decreases the cost.

## 4 METHOD

In § 4.1 below, we explain why, unlike K-Means, the original DP-Means algorithm [Kulis and Jordan, 2012] does not lend itself to parallelization. In that section, we also describe two previous works that tried, with only a partial success, to attack that problem. Next, in § 4.2, we discuss our solution

and the resulting proposed algorithms.

## 4.1 WHY SCALING DP-MEANS IS DIFFICULT

A natural question arises: can the original DP-means algorithm [Kulis and Jordan, 2012] be optimized and parallelized as easily as it was in the K-Means case? Unfortunately, the answer is negative, as we explain below.

When attempting to create a parallel version of DP-Means, the main obstacle is the cluster creation. A naive solution would be to try to simply mimic the steps that are performed when parallelizing the K-Means algorithm: *i.e.*, split the data into $P$ parts, and perform the iteration's main loop (lines 5-10 in Algorithm 2) in parallel. However, when the processing of each part is done, we will (usually) have multiple new clusters and, with a high probability, many of them will overlap with each other. This **over-clustering problem** has two major negative implications. The first and most obvious one is that it harms the results of the clustering. The second ramification of the over-clustering problem is a significant increase of the running time: as the latter grows with $K$, redundant clusters translate directly into a longer running time. Moreover, **a second problem** that stems from the same root cause is that **the efficient computation presented in § 3 for optimizing the distance calculations cannot be done as efficiently as in the K-Means case**: since $K$ (usually) grows during the main iteration, we cannot precompute, before each iteration over the data, all of the $\|\boldsymbol{\mu}_k\|_{\ell_2}^2$ and $\boldsymbol{x}_i^T \boldsymbol{\mu}_k$ values. Rather, we would need to compute these values, on the fly, separately for each new cluster. Consequently, this will deprive us of the benefit of utilizing the maximal efficiency of BLAS.

The remainder of this section reviews two smart existing parallel methods that address the first problem. However, neither of them addresses the second.

**P-DP-Means**, proposed by Pan et al. [2013], splits the data into several parts and then processes them in parallel. The core of that method is that when the calculation of each part is done, the new clusters are not immediately added to the existing clusters; rather, another subroutine, coined 'DPValidate', is called. The additional subroutine consolidates the results from the different parts, and adds the new clusters one by one, as long as the new cluster is distanced by at least $\sqrt{\lambda}$ from all the current existing clusters. However, if a new cluster is not far enough from the existing clusters, then the subroutine will change all the relevant labels to the closest existing cluster. That solution has many merits, and it can indeed reduce the running time drastically without harming the results. However, the 'DPValidate' subroutine itself is serial and slow. This is especially evident in the first iteration of the algorithm, where most of the clusters are added. For example, consider Figure 1 which shows that the first iteration of P-DP-Means is *very* slow, though after that

iteration it converges quite fast. This is also due to the fact that once most of the clusters have been added in the first iteration, one can optimize the distance calculations for all existing clusters for the following iterations, considerably improving the speed of each such iteration.

**DACE** was proposed by Jiang et al. [2017] for clustering extremely-large sequence data in a specific application domain. However, DACE is also fairly easy to adapt to other types of data as well. Jiang et al. [2017] have approached the problem differently from how it was done in P-DP-Means [Pan et al., 2013]. Instead of separating the data into parts and consolidating the results after each iteration, they consolidate only once, running a standard DP-Means algorithm separately on each one of the parts until convergence. Here, the core of the method lies with how the separation into parts is done. Unlike in P-DP-Means, where the partitioning is done using some set heuristic (*e.g.*, it could be random, or according to the data ordering), DACE uses a locality-sensitive hashing approach [Datar et al., 2004] for partitioning the data such that the different parts should have a minimal overlap with each other. This approach has several benefits. First and foremost, it allows the parallelization of DP-Means across the data parts. Second, recall that the runtime grows linearly with $K$. When partitioning the data such that the clusters have a minimal overlap with each other, each part tends to have a low number of clusters. This is in contrast to P-DP-Means, where computations in each part must use all of the clusters. This difference allows DACE a better speedup. DACE, however, has two main drawbacks. The first is that each run of DP-Means has the same optimization problems we have described earlier. The second is that the final result is drastically affected by the initial partitioning and in many cases this leads to the degradation of the results.

## 4.2 THE PROPOSED ALGORITHMS

In § 4.1 we have identified that the main problem with scaling DP-Means is related to cluster creation. This insight leads us to our first proposal: *deferring the cluster creation to the end of the assignment step.* Concretely, when the squared distance between an observation $\boldsymbol{x}_i$ and the center of its nearest cluster exceeds $\lambda$, instead of opening a new cluster, we save the index and distance of that observation in $i_{\max}$ and $d_{\max}$, respectively. We update $i_{\max}$ and $d_{\max}$ whenever we find another observation whose associated distance is larger. Only when the assignment step is complete, and provided that there was at least one observation whose squared distance (from its nearest cluster) exceeds $\lambda$, do we open a new cluster. In which case, that cluster is initialized with the single point whose associated distance was the maximal one across the entire dataset. Next, we continue to update the means of all the existing clusters. We refer to that algorithm, summarized in Algorithm 3, as **DC-DP-Means**

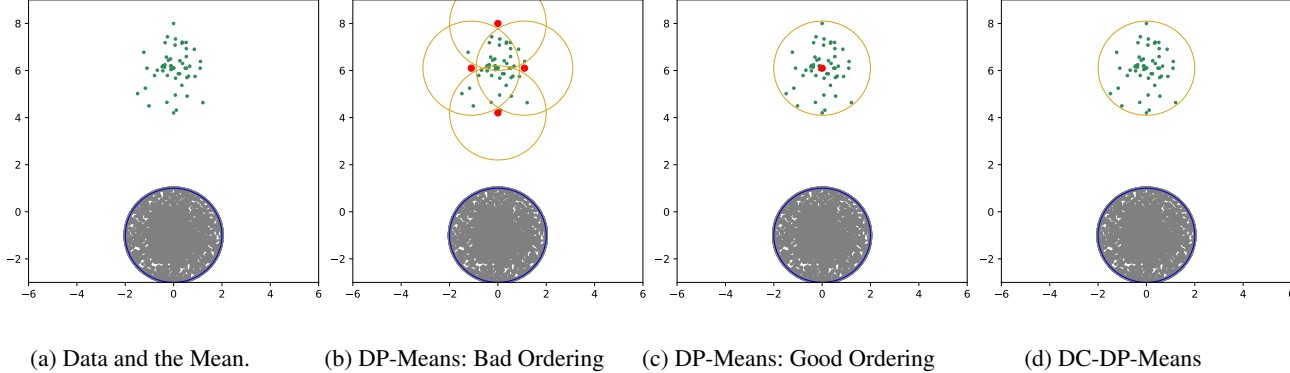

| (a) Data and the Mean. | (b) DP-Means: Bad Ordering | (c) DP-Means: Good Ordering | (d) DC-DP-Means |

Figure 2: DP-Means is more susceptible to over-clustering than PDC-DP-Means. A large number of points are sampled from the bottom cluster (grey) while far fewer ones are sampled from the top cluster (green). Circles of radius $\sqrt{\lambda}$ are drawn around the mean and the instantiated clusters. Red dots mark the first observations visited in DP-Means.

---

**Algorithm 3:** DC-DP-Means

**Input:** $\lambda$
**Data:** $\boldsymbol{X} = (\boldsymbol{x}_i)_{i=1}^N \subset \mathbb{R}^d$

1 $K \leftarrow 1$
2 $\boldsymbol{\mu}_1 \leftarrow \frac{\sum_{i=1}^N \boldsymbol{x}_i}{N}$
3 $(z_i)_{i=1}^N \leftarrow 1$     // init. all labels to 1
4 **while** *Not Converged* **do**
5     $j_{\max} \leftarrow -1$
6     $d_{\max} \leftarrow -1$
7     **for** $i \in \{1, \dots, N\}$ **do**
8         $z_i \leftarrow \arg\min_{k \in \{1,\dots,K\}} \|\boldsymbol{x}_i - \boldsymbol{\mu}_k\|_{\ell_2}^2$
9         **if** $\|\boldsymbol{x}_i - \boldsymbol{\mu}_{z_i}\|_{\ell_2}^2 > d_{\max}$ **then**
10             $j_{\max} \leftarrow i$
11             $d_{\max} \leftarrow \|\boldsymbol{x}_i - \boldsymbol{\mu}_{z_i}\|_{\ell_2}^2$
12     **if** $d_{\max} > \lambda$ **then**
13         $K \leftarrow K + 1$
14         $\boldsymbol{\mu}_K \leftarrow \boldsymbol{x}_{j_{\max}}$
15         $z_{j_{\max}} \leftarrow K$
16     **for** $k = 1 \in \{1, \dots, K\}$ **do**
17         $n_k \leftarrow |\{i : z_i = k\}|$
18         $\boldsymbol{\mu}_k \leftarrow \frac{\sum_{i: z_i = k} \boldsymbol{x}_i}{n_k}$

---

(where DC stands for Delayed Cluster).

*While the delayed cluster creation might seem as a mild change, it has a remarkably-profound threefold effect*: 1) it facilitates the usage of *all* of the K-Means-related optimizations from § 3; 2) DC-DP-Means's results are (trivially) invariant to the ordering of the observations and this is in sharp contrast to the original DP-Means which might severely over-cluster the data due to an unfortunate ordering of the observations (see Figure 2); 3) while DC-DP-Means takes more iterations to converge (since at each iteration at most one cluster can be formed), most of the iterations are

*much* faster (due to, *e.g.*, the fewer clusters and the availability of the aforementioned optimizations), resulting in significantly-short running times.

#### 4.2.1 Convergence Guarantees for DC-DP-Means

The proposed change (from Algorithm 2 to our proposed Algorithm 3) does not break the convergence guarantees of the original DP-Means. In the reassignment step, DC-DP-Means is identical to K-Means as no new clusters are created and the distance between a point and the assigned cluster cannot exceed its pre-reassignment distance. During the cluster creation step, it is guaranteed that $d_{\max}^2 > \lambda$ and thus the added penalty is smaller than $d_{\max}^2$, which is the contribution of the associated observation to the cost function. Finally, during the update of the means, the mean of the observations assigned to a cluster minimizes its squared distance between them and the cluster center. As for an empirical evidence of convergence, see Figure 1.

#### 4.2.2 PDC-DP-Means

Importantly, and by design, DC-DP-Means naturally lends itself to parallelization. It follows that combining the K-Means-related optimization and parallelization steps (described in § 3) together with our proposed delayed cluster creation lets us propose our first *parallel* algorithm, called PDC-DP-Means and summarized in Algorithm 4. Where at first sight Algorithm 4 seems longer and more complex than either Algorithm 2 or Algorithm 3, this is mostly because of the added optimizations we have implemented (and that were enabled by our delayed cluster creation). In other words, while the *parallelizable* Algorithm 3 captures the key conceptual details that allow for massive parallelization and optimizations, the *parallel (and optimized)* Algorithm 4 also contains the needed technical/engineering details. We

**Algorithm 4:** PDC-DP-Means

**Input:** $\lambda, P$

**Data:** $\boldsymbol{X} = \begin{bmatrix} \boldsymbol{x}_1 & \ldots & \boldsymbol{x}_N \end{bmatrix}^T \in \mathbb{R}^{N \times d}$

1  $K \leftarrow 1$

2  $\boldsymbol{\mu}_1 \leftarrow \frac{\sum_{i=1}^{N} \boldsymbol{x}_i}{N}$

3  $(z_i)_{i=1}^{N} \leftarrow 1$     // init. all labels to 1

4  $(\boldsymbol{X}_p)_{p=1}^{P} \leftarrow$ Split $\boldsymbol{X}$ into $P$ parts

5  $(N_p)_{p=1}^{P} \leftarrow$ (#of points in $\boldsymbol{X}_p)_{p=1}^{P}$ // $\boldsymbol{X}_p \in \mathbb{R}^{N_p \times d}$

6  **for** $p \in \{1, \ldots, P\}$ **do in parallel**

7      $\boldsymbol{s}_p \leftarrow (\|\boldsymbol{X}_p[j,:]\|_{\ell_2}^2)_{j=1}^{N_p}$ // $\boldsymbol{s}_p \in \mathbb{R}^{N_p}$

8  **while** *Not Converged* **do**

9      $\boldsymbol{M} \leftarrow \begin{bmatrix} \boldsymbol{\mu}_1 & \ldots & \boldsymbol{\mu}_K \end{bmatrix}^T$     // $\boldsymbol{M} \in \mathbb{R}^{K \times d}$

10      $\bar{\boldsymbol{s}} \leftarrow \begin{bmatrix} \|\boldsymbol{\mu}_1\|_{\ell_2}^2 & \cdots & \|\boldsymbol{\mu}_K\|_{\ell_2}^2 \end{bmatrix}$     // $\bar{\boldsymbol{s}} \in \mathbb{R}^K$

11      $(j_{\max}^p)_{p=1}^{P} \leftarrow -1$   // init. argmax vals

12      $(d_{\max}^p)_{p=1}^{P} \leftarrow -1$   // init. max vals

13      **for** $p \in \{1, \ldots, P\}$ **do in parallel**

14          $\boldsymbol{M}_p = \begin{bmatrix} \boldsymbol{\mu}_{1,p} & \ldots & \boldsymbol{\mu}_{K,p} \end{bmatrix}^T \leftarrow \boldsymbol{0}_{K \times d}$

15          $(n_{k,p})_{k=1}^{K} \leftarrow 0$     // init. counts

16          $\boldsymbol{D}_p \leftarrow -2\boldsymbol{X}_p \boldsymbol{M}^T + \underbrace{\begin{bmatrix} \bar{\boldsymbol{s}} & \ldots & \bar{\boldsymbol{s}} \end{bmatrix}}_{K \text{ copies of } \bar{\boldsymbol{s}}}$  //

        $\boldsymbol{D}_p \in \mathbb{R}^{N_p \times K}$

17          $\boldsymbol{z}_p \leftarrow$ row-wise argmin $(\boldsymbol{D}_p)$

18          **for** $j \in \{1, \ldots, N_p\}$ **do**

19              $k \leftarrow \boldsymbol{z}_p[j]$

20              $\boldsymbol{M}_p[k,:] \leftarrow \boldsymbol{M}_p[k,:] + \boldsymbol{X}_p[j,:]$

21              $n_{k,p} \leftarrow n_{k,p} + 1$

22              **if** $\boldsymbol{D}_p[j,k] + \boldsymbol{s}_p[j] > \max(d_{\max}^p, \lambda)$ **then**

23                  $d_{\max}^p \leftarrow \boldsymbol{D}_p[j,k] + \boldsymbol{s}_p[j]$

24                  $j_{\max}^p \leftarrow j$

25      $p_{\max} \leftarrow \arg\max_p (d_{\max}^p)_{p=1}^{P}$

26      $\begin{bmatrix} n_1 & \ldots & n_K \end{bmatrix} \leftarrow \begin{bmatrix} \sum_{p=1}^{P} n_{1,p} & \ldots & \sum_{p=1}^{P} n_{K,p} \end{bmatrix}$

27      $\boldsymbol{M} \leftarrow \sum_{p=1}^{P} \boldsymbol{M}_p$

28      **if** $p_{max} \neq -1$ **then**

29          $j \leftarrow j_{\max}^{p_{\max}}$

30          $k \leftarrow \boldsymbol{z}_{p_{\max}}[j]$

31          $\boldsymbol{M}[k,:] \leftarrow \boldsymbol{M}[k,:] - \boldsymbol{X}_p[j,:]$

32          $n_k \leftarrow n_k - 1$

33          $K \leftarrow K + 1$

34          $\boldsymbol{z}_{p_{\max}}[j] \leftarrow K$

35          $\boldsymbol{M}[K,:] \leftarrow \boldsymbol{X}_p[j,:]$

36          $n_K \leftarrow 1$

37      **for** $k \in \{1, \ldots, K\}$ **do in parallel**

38          $\boldsymbol{\mu}_k \leftarrow \boldsymbol{M}[k,:]/n_k$

now explain the details.

In *lines 4-5*, we split the data into $P$ parts (where $P$ is the number of available computer processes), where $\boldsymbol{X}_p$ denotes the observations in part $p$. Throughout the algorithm, we use $j$ as the running index within a part while $i$ is used as the running index within the entire dataset. In *lines 6-7* we

pre-calculate $\boldsymbol{s}_p$, which stands for the squared norms of the points, for each of the $P$ parts. This $\boldsymbol{s}_p$ will be used later for comparing distances. In each of the iterations of the *while* loop (*line 8*) we initially calculate the squared norms $\bar{\boldsymbol{s}}$ of the cluster centers (to be used by all the $P$ processes), and initialize auxiliary variables $(j_{\max}^p)_{p=1}^{P}, (d_{\max}^p)_{p=1}^{P}$ to hold the index and distance of the observation with the maximal distance (from the existing clusters) within each part.

We then process the data in parallel, where for each part $p$ of the data we initialize several auxiliary variables: $\boldsymbol{M}_p$, for storing the sum of the observations (in part $p$) assigned to each cluster, and $(n_{k,p})_{k=1}^{K}$, for storing the number of observations (in part $p$) assigned to each cluster. In $\boldsymbol{D}_p$ we store the computed distances between the observations in $\boldsymbol{X}_p$ and each of the $K$ centers. Taking the argmin for each row of $\boldsymbol{D}_p$, we then procure the labels for the data in $\boldsymbol{X}_p$.

Iterating over the observations, in *lines 18-24* we update the contribution of each observation to the cluster centers, and update both $\boldsymbol{M}_p$ and $(n_{k,p})_{k=1}^{K}$ accordingly. While doing so we also check for the maximal distance between an observation and its assigned center in this data part, updating $j_{\max}^p$ and $d_{\max}^p$ accordingly. Finally, in *lines 28-36* we create a new cluster if necessary, using the farthest observation across all the $P$ data parts, but only provided that the squared distance exceeds $\lambda$. We then aggregate the results from the parts and use them to update the cluster centers.

To declare convergence, we may choose one of three possible criteria: 1) no label switching between iterations; 2) measuring the difference in the cost function values between two iterations, and declaring convergence if it is smaller than some pre-defined tolerance value; 3) measuring the distances between the centers in two consecutive iterations, and declaring convergence if they are smaller than some pre-defined tolerance value. Criterion 1 is the strongest, and usually takes longer to achieve than the others. While criteria 1 and 3 cannot be met if a new cluster has been introduced in the latest iteration, criterion 2 can be met even if there is a cluster which existed only in the last iteration.

### 4.2.3 DACE-PDC-DP-Means

Recall that a key property in DACE is that the vanilla DP-Means is used as a subroutine: the data is partitioned according to the desired parallelism, and the DP-Means subroutine is used independently on each of the parts. We now propose replacing that subroutine with our proposed PDC-DP-Means. This has several benefits over the original DACE. First, we utilize all the pros of using PDC-DP-Means, which include both parallelism and, which is more important in this case, optimizations of the different calculations. Second, this overcomes the main drawback of PDC-DP-Means, where the delayed cluster creation can slow down convergence on datasets presenting a very high number of clusters.

Splitting the calculation into several parts (especially if there is little or no overlap of clusters between the parts – as is the case with DACE), solves this problem. Moreover, it also gives the user control over the pace at which $K$ grows. That is, the user can choose between maximal data-partition parallelism for an increased cluster-creation speed and a coarser data partition that allows the PDC-DP-Means subroutine to employ additional parallelism within each part.

### 4.2.4 MiniBatch PDC-DP-Means

The delayed creation also lets us extend PDC-DP-Means to a Mini-Batch setting. The transition is similar to the transition from K-Means to Mini-Batch K-Means [Sculley, 2010]. Instead of evaluating the entire dataset at once, we randomly sample a subset (called a mini-batch) $X_b$ of size $b$ from the dataset and run a PDC-DP-Means iteration on it, *without updating the centers*. We parallelize the processing of $X_b$ across the available cores. In each batch $X_b$ we cache both the index and the distance of the most distances observation $x_j$, and if that observation is at a distance of at least $\sqrt{\lambda}$ from its nearest cluster, we instantiate a new cluster, centered at $x_j$. Unlike in PDC-DP-Means, however, here we do not recalculate the cluster centers in each iteration; rather, instead we take a step towards the observations assigned to the cluster, using the following (gradient-based) formula,

$$\boldsymbol{\mu}_k \leftarrow \left(1 - \frac{1}{n_k}\right)\boldsymbol{\mu}_k + \frac{1}{n_k}\boldsymbol{x}_j, \qquad (4)$$

where $\boldsymbol{\mu}_k$ is the current cluster center, $\boldsymbol{x}_j$ is the new observation assigned to cluster $k$, and $n_k$ is the total number of observations assigned to cluster $k$, including $\boldsymbol{x}_j$. We present the full algorithm in Algorithm 5. To determine convergence we need to modify some of the aforementioned criteria. While the criterion regarding the distance between the centers can remain the same (though the *tolerance* value needs to be adjusted), the other two are no longer applicable as they require the entire dataset to be processed. Instead (and as is done in MiniBatch K-Means), we can evaluate these two criteria on a pre-defined validation set.

**Online Setting.** As in the original MiniBatch K-Means, our algorithm also supports an online setting, where the main iteration (*lines 6-33*) is executed not on some sample from the dataset, but on the current available data. When new data arrives, we process it in the main iteration. Thus, there is no need to store the previously-seen data at any point in time.

## 5 EXPERIMENTS AND RESULTS

To validate the utility of our methods, we have compared them with various methods in different settings. All of our experiments were run on an Ubuntu 20.4 machine with 64GB RAM and Intel® Core™ i9-11900K Processor.

---

**Algorithm 5:** MiniBatch PDC-DP-Means

**Input:** $\lambda, P, b$
**Data:** $\boldsymbol{X} = \begin{bmatrix} \boldsymbol{x}_1 & \dots & \boldsymbol{x}_N \end{bmatrix}^T \in \mathbb{R}^{N \times d}$

1   $K \leftarrow 1$
2   $\mu_1 \leftarrow$ Random point from $\boldsymbol{X}$
3   $n_1 \leftarrow 1$
4   **while** *Not Converged* **do**
5      $\boldsymbol{X}^b \leftarrow b$ random points from $\boldsymbol{X}$
6      $(\boldsymbol{X}_p)_{p=1}^P \leftarrow$ Split $\boldsymbol{X}^b$ into $P$ parts
7      $(N_p)_{p=1}^P \leftarrow$ (#of pts in $\boldsymbol{X}_p)_{p=1}^P$ // $\boldsymbol{X}_p \in \mathbb{R}^{N_p \times d}$
8      **for** $p \in \{1, \dots, P\}$ **do in parallel**
9         $\boldsymbol{s}_p \leftarrow (\|\boldsymbol{X}_p^b[j,:]\|_{\ell_2}^2)_{j=1}^{N_p}$ // $\boldsymbol{s}_p \in \mathbb{R}^{N_p}$
10      $\boldsymbol{M} \leftarrow \begin{bmatrix} \boldsymbol{\mu}_1 & \dots & \boldsymbol{\mu}_K \end{bmatrix}^T$    // $\boldsymbol{M} \in \mathbb{R}^{K \times d}$
11      $\bar{\boldsymbol{s}} \leftarrow \begin{bmatrix} \|\boldsymbol{\mu}_1\|_{\ell_2}^2 & \cdots & \|\boldsymbol{\mu}_K\|_{\ell_2}^2 \end{bmatrix}$    // $\bar{\boldsymbol{s}} \in \mathbb{R}^K$
12      $(j_{\max}^p)_{p=1}^P \leftarrow -1$   // init. argmax vals
13      $(d_{\max}^p)_{p=1}^P \leftarrow -1$     // init. max vals
14      **for** $p \in \{1, \dots, P\}$ **do in parallel**
15         $\boldsymbol{D}_p \leftarrow -2\boldsymbol{X}_p\boldsymbol{M}^T + \bar{\boldsymbol{s}}$    // $\boldsymbol{D}_p \in \mathbb{R}^{N_p \times K}$
16         $\boldsymbol{z}_p \leftarrow$ row-wise argmin $(\boldsymbol{D}_p)$
17         **for** $j \in \{1, \dots, N_p\}$ **do**
18            **if** $\boldsymbol{D}_p[j, k] + \boldsymbol{s}_p[j] > \max(d_{\max}^p, \lambda)$ **then**
19               $d_{\max}^p \leftarrow \boldsymbol{D}_p[j, k] + \boldsymbol{s}_p[j]$
20               $j_{\max}^p \leftarrow j$
21      $p_{\max} \leftarrow \arg\max_p(d_{\max}^p)_{p=1}^P$
22      **if** $p_{max} \neq -1$ **then**
23         $j \leftarrow j_{\max}^{p_{\max}}$
24         $K \leftarrow K + 1$
25         $\boldsymbol{z}_{p_{\max}}[j] \leftarrow K$
26         $n_K \leftarrow 0$
27         $\boldsymbol{\mu}_K \leftarrow \boldsymbol{X}_p[j,:]$
28      $\boldsymbol{z} \leftarrow (\boldsymbol{z}_1, \dots, \boldsymbol{z}_P)$
29      **for** $k \in \{1, \dots, K\}$ **do in parallel**
30         $X_k^b \leftarrow \{\boldsymbol{x}_j \in \boldsymbol{X}^b : \boldsymbol{z}[j] = k\}$
31         **for** $x_j \in X_k^b$ **do**
32            $n_k \leftarrow n_k + 1$
33            $\boldsymbol{\mu}_k \leftarrow (1 - \frac{1}{n_k})\boldsymbol{\mu}_k + \frac{\boldsymbol{x}_j}{n_k}$

---

**Methods and Implementations.** The implementation of our proposed methods is in Python and Cython, and we have integrated it within the code base of Scikit-learn [Pedregosa et al., 2011]. When writing the implementation we had both efficiency and accessibility in mind. In particular, we exploit Scikit-learn's efficient codebase, while making the use of our code an easy "drop-and-replacement". That is, a user that previously used Scikit-learn's K-Means or MiniBatch K-Means, can now simply change to our code using the same interface (except that instead of passing $K$ as a parameter the user will pass $\lambda$). For the vanilla DP-Means we have used the publicly-available *R* package *'maotai'*. As public implementation of DACE [Jiang et al., 2017] is aimed towards RNA sequence data, we have created our version of it which is more general and can handle any

Table 1: Comparing running time and NMI of different algorithms on various datasets. Our proposed methods uniformly have better results than the other DP-Means variants, and in most cases, better than the K-Means variants as well. Note that the parametric methods (marked by $^\dagger$), which had to be given the true $K$ so they had an unfair advantage, are included here only for completeness. The important comparison, however, is between the nonparametric ones.

| Dataset | 2D Gaussian | | 10D Gaussian | | MNIST | | ImageNet100 | | ImageNet1K | |
|---|---|---|---|---|---|---|---|---|---|---|
| Method | NMI | Time [sec] | NMI | Time [sec] | NMI | Time [sec] | NMI | Time [sec] | NMI | Time [sec] |
| K-Means$^\dagger$ | .872 ± .002 | 1.47 ± 0.01 | .634 ± .003 | 1.35 ± 0.0 | .492 ± .005 | 0.12 ± 0.00 | .770 ± .001 | 1.53 ± 0.06 | .736 ± .000 | 198 ± 17 |
| MiniBatch K-Means$^\dagger$ | .875 ± .004 | 0.27 ± 0.11 | .632 ± .013 | 0.04 ± 0.02 | .451 ± .025 | 0.15 ± 0.04 | .762 ± .002 | 0.29 ± 0.00 | .727 ± .000 | 4.97 ± 0.28 |
| DP-Means | .883 ± .002 | 865 ± 9 | .666 ± .001 | 459 ± 63 | .534 ± .001 | 204 ± 57 | .765 ± .001 | 205 ± 87 | N/A | N/A |
| DACE | .890 ± .003 | 35.4 ± 6.5 | .648 ± .003 | 9.92 ± 1.19 | .506 ± .003 | 4.86 ± 0.64 | .730 ± .003 | 34.5 ± 3.9 | .720 ± .002 | 8501 ± 613 |
| P-DP-Means | .884 ± .002 | 117 ± 1 | .686 ± .007 | 37.1 ± 7.33 | .532 ± .000 | 17.5 ± 0.8 | .765 ± .001 | 8.53 ± 0.65 | .729 ± .000 | 424 ± 24 |
| PDC-DP-Means (Ours) | **.891 ± .006** | 3.55 ± 0.25 | **.713 ± .000** | 10.8 ± 1 | **.540 ± .002** | 0.96 ± 0.03 | **.767 ± .000** | 2.47 ± 0.32 | **.734 ± .000** | 1232 ± 66 |
| DACE - PDC-DP-Means (Ours) | .888 ± .006 | 9.96 ± 2.65 | .663 ± .012 | 2.38 ± 0.12 | .498 ± .001 | 0.51 ± 0.01 | .749 ± .003 | 3.73 ± 0.20 | .731 ± .005 | 123 ± 17 |
| MiniBatch PDC-DP-Means (Ours) | .882 ± .010 | **1.07 ± 0.16** | .645 ± .017 | **0.33 ± 0.01** | .501 ± .004 | **0.43 ± 0.15** | .758 ± .006 | **0.39 ± 0.23** | .728 ± .000 | **12.9 ± 1.07** |

data type. Our pure DACE version uses the aforementioned 'maotai' package for the DP-Means subroutine, while in our DACE+DCP version we have simply changed the subroutine to our proposed PDP-DP-Means. For P-DP-Means [Pan et al., 2013] there is no publicly-available implementation, so we have created our own efficient implementation of it, written in Python and utilizing Scikit-learn's efficient Cython subroutines. Finally, for K-Means and MiniBatch K-Means, we have used the available optimized Scikit-learn implementations.

**Datasets.** We have used several datasets: a synthetic 2D Dataset, with $N = 10^6$ points, sampled from a 50-component Gaussian Mixture Model (GMM); a synthetic 10D Dataset, with $N = 10^5$ points, sampled from a 20-component GMM; MNIST [LeCun, 1998] handwritten digits dataset with dimensionality reduced to 16 using Principal Component Analysis (PCA); ImageNet100 [Deng et al., 2009] (a subset of the entire ImageNet dataset), which consists of 125K images that belong to 100 classes from the entire ImageNet, where we also used SWAV [Caron et al., 2020] to extract features from the images, followed by PCA to reduce the dimensionality of the features to 64; ImageNet1K [Deng et al., 2009], which is the the full IL-SRVC2012 dataset train set, containing 1.2M images from 1000 classes and where we again used SWAV followed by PCA to reduce the dimensionality, this time to 128. We emphasize that the dimensionality reduction was done mostly for the benefit of the other methods (our methods, which scale better, can handle higher dimensions).

**Evaluation.** We have split the data into Train-Validation-Test sets, in proportions of $0.9, 0.02, 0.08$, respectively. In order to evaluate the results of the clustering, we have used a model-independent metric, the Normalized Mutual Information (NMI) score. While all DP-Means variants share the same cost function, given the same $\lambda$, the expected clustering results differ by a lot: DACE and MiniBatch PDC-DP-Means will usually output a higher number of clusters than DP-Means and P-DP-Means, while the latter two usually output a higher number of cluster than PDC-DP-Means. As

such, we have optimized the $\lambda$ value independently for each of the models, using the validation set, setting NMI as the target function for the optimization and using [Knysh and Korkolis, 2016] as the optimizer. The full results for this setting appear in Table 1. From the table, it is observable that PDC-DP-Means outperforms all DP-Means variants in terms of NMI and that in most cases it outperforms even the parametric methods, which had to be given the true $K$. In terms of running time, our MiniBatch PDC-DP-Means is always the fastest DP-Means-related method, usually by a very large margin, despite having only a slight reduction in the quality of the results. Also, in almost all cases it outperforms the only-slightly faster MiniBatch K-Means, in terms of NMI score. An interesting observation is that in the ImageNet1K case (where the true $K$ is high: 1000), our proposed PDC-DP-Means is slower than P-DP-Means. This is the only case where the delayed cluster creation harms the running time due to the large number of clusters. P-DP-Means instantiates most of the clusters in the first few iterations, and this enables it optimizing the distance calculations for most of the clusters very early. However, both our MiniBatch PDC-DP-Means and our DACE-PDC-DP-Means do not suffer from the large $K$: while both use delayed cluster creation, the MiniBatch PDC-DP-Means does so after every MiniBatch, while DACE-PDC-DP-Means splits the data into parts and thus can create multiple clusters at the same time. As evident by the results, both methods converge much faster than PDC-DP-Means, with a similar quality of results.

**Comparing with Nonparametric Algorithms.** So far we have focused on comparisons with either DP-Means, K-Means, or their variants. However, there are other non-parametric clustering algorithms (with existing efficient implementations) that are unrelated to DP-Means. In particular, we have compared with the following popular algorithms: DBSCAN [Ester et al., 1996], MeanShift [Comaniciu and Meer, 2002], Agglomerative Clustering [Maimon and Rokach, 2005] and OPTICS [Ankerst et al., 1999]. All the above implementations are available in Scikit-learn [Pedregosa et al., 2011]. To compare with those algorithms, we

Table 2: Comparison with nonparametric clustering algorithms

| Dataset | 2D Gaussian | | MNIST | | ImageNet100 | |
|---|---|---|---|---|---|---|
| Method | NMI | Time [sec] | NMI | Time [sec] | NMI | Time [sec] |
| DBSCAN | $.69 \pm .00$ | $0.58 \pm 0.03$ | $.35 \pm .00$ | $0.96 \pm 0.05$ | $.580 \pm .00$ | $94 \pm 14$ |
| MeanShift | $.74 \pm .00$ | $125 \pm 2.0$ | $.43 \pm .00$ | $781 \pm 11$ | N/A | N/A |
| Agglomerative Clustering | $.82 \pm .00$ | $37.7 \pm 0.3$ | $.50 \pm .00$ | $74.4 \pm 1.9$ | N/A | N/A |
| OPTICS | $.75 \pm .00$ | $27.2 \pm 0.21$ | $.02 \pm .00$ | $49.3 \pm 0.46$ | N/A | N/A |
| PDC-DP-Means (Ours) | $\mathbf{.83 \pm .01}$ | $0.07 \pm 0.00$ | $\mathbf{.51 \pm .00}$ | $0.84 \pm 0.26$ | $\mathbf{.76 \pm .00}$ | $1.9 \pm 0.29$ |
| MiniBatch PDC-DP-Means (Ours) | $.82 \pm .03$ | $\mathbf{0.04 \pm 0.02}$ | $.45 \pm .01$ | $\mathbf{0.19 \pm 0.04}$ | $.74 \pm .00$ | $\mathbf{0.52 \pm 0.00}$ |

have used smaller versions of the 2D GMM and MNIST datasets, the former with only $50K$ observations sampled from 20 2D Gaussians, while the latter is MNIST train set with dimensionality reduced to 8 via PCA. In addition to those 2 datasets, we have used the previously-discussed ImageNet100 for comparing with DBSCAN, the only other nonparametric method which could scale to a dataset of such a size. We note that while our method (and some of the other methods) can gracefully handle very large datasets, MeanShift's [Comaniciu and Meer, 2002] runtime and Agglomerative Clustering's [Maimon and Rokach, 2005] memory consumption makes them impractical for large datasets. Table 2 summarizes the results, showing that our approach not only outperforms the others in terms of clustering results but also does it in a fraction of the time. Note that for each method, we have optimized its parameters using black-box optimization [Knysh and Korkolis, 2016] on the data test set. This is in contrast to the previous experiment where we have used the validation set. Here, however, some algorithms (*e.g.*, DBSCAN) cannot predict the labels of new samples. Thus, we have evaluated the performance on the clustering results of the train set (hence the discrepancy between the ImageNet100 results here of PDC-DP-Means and the Minibatch and their counterparts in Table 1).

## 6 CONCLUSION

In this paper we have focused on the practical aspects of parallelizing DP-Means. We have examined previous attempts at that goal and proposed several algorithms which are parallel, highly efficient, and usually achieve better clustering results than their counterparts. Our main contribution, the PDC-DP-Means, has one key limitation: as the number of clusters can only increase by one in each iteration, in data with a large $K$ (*e.g.*, ImageNet1K), this can lead to a large number of iterations. However, our other proposed algorithms, DACE-PDC-DP-Means and MiniBatch PDC-DP-Means, offer a remedy in those cases as the number of clusters can increase very fast, as is evident by our results. To summarize, our recommendation is to use PDC-DP-Means for datasets where one may expect to find only a moderate $K$. This will usually yield the best results. If $K$ is expected to be large, using either DACE-PDC-DP-Means or MiniBatch PDC-DP-Means is preferred (despite the mild

drop in the quality of the results) due to the major speedups.

## Acknowledgements

This work was supported by the Lynn and William Frankel Center at BGU CS, by the Israeli Council for Higher Education via the BGU Data Science Research Center, and by Israel Science Foundation Personal Grant #360/21. O.D. was also funded by the Jabotinsky Scholarship from Israel's Ministry of Technology and Science, and by BGU's Hi-Tech Scholarship.

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
