# OpenReview forum: "Revisiting DP-Means: Fast Scalable Algorithms via Parallelism and Delayed Cluster Creation"
_auai.org/UAI/2022/Conference — UAI 2022 Poster_

### Official Review · Reviewer_61xf · 2022-03-24

**Q2(1) Originality/Novelty:** 3
**Q2(2) Significance/Impact:** 3
**Q2(3) Correctness/Technical Quality:** 3
**Q2(6) Clarity Of Writing:** 3
**Q6 Overall Score:** 7
**Q8 Confidence In Your Score:** 1

**Q1 Summary And Contributions:**

This paper proposes a new parallel version of DP-Means: PDC-DP-Means, with 2 extensions. The core idea is to postpone the cluster creation for later decision at the end of the assignment step. The experiments show that the proposed methods can achieve better NMI with much less time compared to the baselines.

**Q2 Assessment Of The Paper:**

More detailed information regarding each of these aspects is given below:

**Q2(4) Quality Of Experiments (Optional):**

3: Good: The experimental evaluation is adequate, and the results convincingly support the main claims.

**Q2(5) Reproducibility:**

3: Good: Key resources (e.g., proofs, code, data) are available and key details (e.g., proofs, experimental setup) are sufficiently well-described for competent researchers to confidently reproduce the main results.

**Q3 Main Strengths:**

1. The main idea is neat and clean. The proposed methods are technically sound and make sense to me.
2. The experiments show that the proposed methods can achieve better NMI with much less time compared to the baselines.

**Q4 Main Weakness:**

1. Maybe K-Means should also be included as a baseline.

**Q5 Detailed Comments To The Authors:**

In overall I think the main idea is neat and clean, and makes sense to me. The experiments show that the proposed methods can achieve better NMI with much less time compared to the baselines.
Since K-Means is relatively easier to be parallelized, I wonder whether the SOTA parallel K-Means algorithm should be included as a baseline.
I also wonder whether there is typically any theoretical analysis for DP-Means.

**Q7 Justification For Your Score:**

I think the main idea is good and interesting. The experiment results look quite good to me. However, I'm not very familiar with this specific topic.

**Q9 Complying With Reviewing Instructions:**

1: Yes.

---

### Official Review · Reviewer_cdTJ · 2022-04-04

**Q2(1) Originality/Novelty:** 2
**Q2(2) Significance/Impact:** 3
**Q2(3) Correctness/Technical Quality:** 4
**Q2(6) Clarity Of Writing:** 4
**Q6 Overall Score:** 8
**Q8 Confidence In Your Score:** 3

**Q1 Summary And Contributions:**

The paper studies the problem of improving the efficiency of DP-Means through parallelization.
The paper introduces several algorithms centred around the idea of delayed cluster creation, which show effective performance.

**Q2 Assessment Of The Paper:**

More detailed information regarding each of these aspects is given below:

**Q2(4) Quality Of Experiments (Optional):**

3: Good: The experimental evaluation is adequate, and the results convincingly support the main claims.

**Q2(5) Reproducibility:**

3: Good: Key resources (e.g., proofs, code, data) are available and key details (e.g., proofs, experimental setup) are sufficiently well-described for competent researchers to confidently reproduce the main results.

**Q3 Main Strengths:**

The paper introduces a simple but effective idea of delayed clustering
The paper is very well written
The experiments clearly validate the main claim


**Q4 Main Weakness:**

I cannot identify any major weakness

**Q5 Detailed Comments To The Authors:**

I have enjoyed reading this paper. I am not an expert on clustering, so I am judging the merit of this paper based on whether I have learned something new while reading it.

Originality: the core idea is quite simple, and that is the strength of the paper. The idea itself is a natural extension, but definitely not a trivial one. Due to its simplicity and 'naturalness', it is difficult to deem it ground-breaking, but it certainly introduces ideas that can impact the rest of the field.


Significance: The presented idea is very general and it applies to any clustering algorithm. The algorithms are effective and thus I do expect them to have an impact on the field. Perhaps a minor weakness here is that the proposed idea is not really applicable to other parts of AI, but most papers are like that anyways.


Technical quality: the idea is simple and well-executed. The technical advances focus on improving the efficiency of various parts of the algorithm, and I have not been identified a wrong assumption or result .

Experiments: the experiments are sufficient to validate the method. The experiments involve both synthetic and real-world data, chosen so that the improvement of the method is possible to clearly verify. The method is also compared to a variety of existing algorithms and clearly outperforms them.

Reproducibility: the experiments and algorithms are explained to sufficient detail.

Clarity of writing: this paper is exceptionally well written. I have not worked on clustering for a few years, and never on the non-parametric ones, but the paper is very easy to follow. The analysis of the weaknesses is very well written and opens the door to the contributions of the paper. The algorithm itself is presented well clearly, with the consequence of each decision clearly elaborated. One improvement is still possible -- when describing the main algorithms, it would have been great if the authors explained the algorithm by separating the conceptual from engineering advances.

**Q7 Justification For Your Score:**

My score is high because the contribution is clear, simple and very effective, the experiments sufficiently validate the claims, and the paper is exceptionally well written

**Q9 Complying With Reviewing Instructions:**

1: Yes.

---

### Official Review · Reviewer_uwLV · 2022-04-13

**Q2(1) Originality/Novelty:** 3
**Q2(2) Significance/Impact:** 3
**Q2(3) Correctness/Technical Quality:** 3
**Q2(6) Clarity Of Writing:** 2
**Q6 Overall Score:** 6
**Q8 Confidence In Your Score:** 2

**Q1 Summary And Contributions:**

This paper designs a new parallel algorithm (called PDC-DP-Means) for the DP-mean problem. The authors propose to use a delayed cluster creation strategy, instead of creating a new cluster whenever the creation criterion is satisfied. This strategy enables the algorithm to be run in parallel, thus speedup the clustering process. Experimental results on several datasets show the effectiveness and efficiency of PDC-DP-Means.

**Q2 Assessment Of The Paper:**

More detailed information regarding each of these aspects is given below:

**Q2(4) Quality Of Experiments (Optional):**

3: Good: The experimental evaluation is adequate, and the results convincingly support the main claims.

**Q2(5) Reproducibility:**

3: Good: Key resources (e.g., proofs, code, data) are available and key details (e.g., proofs, experimental setup) are sufficiently well-described for competent researchers to confidently reproduce the main results.

**Q3 Main Strengths:**

1.	This paper proposes a novel efficient algorithm for DP-Means problem.
2.	The delayed cluster creation idea is critical to enable the algorithm parallelizable.
3.	The empirical convergent speed of PDC-DP-Means shown in Figure 1 is remarkable, which verifies the major contribution of this paper.

**Q4 Main Weakness:**

1.	Algorithms 3 and 4 are difficult to follow, adding an example may help explanation.
2.	Although convergence properties are discussed and verified empirically, rigorously analyzing the convergence of algorithms will be better.

**Q5 Detailed Comments To The Authors:**

1.	Step 16 in Algorithm 3: X_p M^\top is a matrix, but \bar{s} is a vector (step 10).
2.	Section 2 and Section 3 can be merged and reorganized.

**Q7 Justification For Your Score:**

I am not an expert in this topic. The proposed algorithm is an important contribution for DP-Means problem in terms of efficiency, especially in big-data scenarios. The delayed cluster creation idea is intuitive. The main contributions are verified by experimental results. As mentioned, adding some theoretical justification can improve the paper and its impact.

**Q9 Complying With Reviewing Instructions:**

1: Yes.

---

### Decision · Program_Chairs · 2022-05-15

**Decision:**

Accept (Poster)

**Comment:**

Meta Review: n this paper, the authors study the problem of improving the efficiency of DP-Means through parallelization. The reviewers consider that this paper is novel and the experiments verify the major contribution of this paper.